# Exploring the Therapeutic Potential of Bromelain: Applications, Benefits, and Mechanisms

**DOI:** 10.3390/nu16132060

**Published:** 2024-06-28

**Authors:** Urna Kansakar, Valentina Trimarco, Maria V. Manzi, Edoardo Cervi, Pasquale Mone, Gaetano Santulli

**Affiliations:** 1Department of Medicine, Division of Cardiology, Wilf Family Cardiovascular Research Institute, Fleischer Institute for Diabetes and Metabolism (*FIDAM*), Albert Einstein College of Medicine, New York, NY 10461, USA; 2Department of Neuroscience, Reproductive Sciences and Dentistry, Federico II University, 80131 Naples, Italy; 3Department of Advanced Biomedical Sciences, Federico II University Hospital, 80131 Naples, Italy; 4Vein Clinic, University of Brescia, 25100 Brescia, Italy; 5Department of Medicine and Health Sciences “Vincenzo Tiberio”, University of Molise, 86100 Campobasso, Italy; 6Casa di Cura “Montevergine”, 83013 Avellino, Italy; 7Department of Molecular Pharmacology, Einstein Institute for Aging Research, Einstein-Mount Sinai Diabetes Research Center (*ES-DRC*), Einstein Institute for Neuroimmunology and Inflammation (*INI*), Albert Einstein College of Medicine, New York, NY 10461, USA

**Keywords:** age, ananain, ananas, antioxidants, bromelain, COVID-19, dermatology, enzyme, fruit, immune system, inflammation, oxidative stress, pineapple, protease, wound healing

## Abstract

Bromelain is a mixture of proteolytic enzymes primarily extracted from the fruit and stem of the pineapple plant (*Ananas comosus*). It has a long history of traditional medicinal use in various cultures, particularly in Central and South America, where pineapple is native. This systematic review will delve into the history, structure, chemical properties, and medical indications of bromelain. Bromelain was first isolated and described in the late 19th century by researchers in Europe, who identified its proteolytic properties. Since then, bromelain has gained recognition in both traditional and modern medicine for its potential therapeutic effects.

## 1. Introduction

The history of bromelain dates back to the ancient civilizations of South America, where the pineapple plant (*Ananas comosus*) is native. Indigenous peoples in Central and South America, particularly in regions like the Amazon rainforest and the Caribbean, used various parts of the pineapple plant for medicinal purposes, including treating digestive issues, reducing inflammation, and healing wounds. The modern history of bromelain began in the late 19^th^ century, when researchers started to explore the potential therapeutic properties of pineapple enzymes [1,2,3,4,5,6]. 

Originally, bromelain was exclusively derived from Hawaiian pineapple stems, but it is now also manufactured in Taiwan, Thailand, Brazil, and Puerto Rico. However, the variability in the commercially produced product and its diverse ingredients have hindered successful development. While pineapple bromelain has found commercial applications as a meat-tenderizing enzyme and a nutraceutical, efforts have been made to explore its potential for pharmaceutical use (Figure 1). Yet, the intricate nature of bromelain’s active components has posed some limitations on pharmaceutical research [7,8,9]. 

The history of bromelain spans centuries, from its traditional use by indigenous peoples to its modern industrial production and medical applications. Ongoing research continues to uncover new insights into its therapeutic potential and broaden its range of uses in various fields.

## 2. Chemical Properties 

Bromelain is a complex mixture of proteolytic enzymes, including various proteases, such as stem bromelain, fruit bromelain, and ananain [10,11]. These enzymes belong to the cysteine protease family and exhibit different substrate specificities and optimal pH ranges [11,12,13,14]. The composition of bromelain can vary depending on factors such as the source of extraction and processing methods [15,16,17,18]. In more detail, bromelain activity occurs between pH 3 and 7. Beyond this value, it declines progressively, causing decreased absorption at higher pH values. Notably, several studies demonstrated that the proteolytic enzymes constituting bromelain mixture are absorbed from the gastrointestinal tract in an intact and functional form then available for intestinal absorption [19,20]. 

For the abovementioned reasons, bromelain is completely absorbed in any form and has no intestinal degradation problems. 


*Fruit Bromelain vs. Stem Bromelain*


EC 3.4.22.33 (Fruit Bromelain) and EC 3.4.22.32 (Stem Bromelain) are both types of bromelain, a group of proteolytic enzymes derived from *Ananas comosus* that belong to the peptidase family C1 [21,22,23]. 

Data adapted from the BRaunschweig ENzyme Database (BRENDA), a collection of enzyme functional information available to the scientific community free of charge and maintained by the Leibniz Institute DSMZ as part of the Digital Diversity project, show that, while the two forms share similarities in their biochemical properties and functions, there are notable differences between them [24,25,26,27,28,29,30], as depicted in Table 1; they both belong to peptidase family C1 (papain family). Another cysteine endopeptidase with similar action on small molecule substrates, pinguinain (formerly EC 3.4.99.18), is obtained from the related plant, *Bromelia pinguin*, but pinguinain differs from fruit and stem bromelain in being inhibited by chicken cystatin [31,32]. 

## 3. Biological Effects

Bromelain exerts its therapeutic effects through a multifaceted mechanism of action, which contribute to the anti-inflammatory, analgesic, antiangiogenic, and antioxidant properties of bromelain, making it a promising candidate for the treatment of various inflammatory and oxidative stress-related disorders; its diverse biological effects depend on multiple mechanisms of action, including proteolytic activity, anti-inflammatory and immunomodulatory effects, fibrinolytic activity, antioxidant properties, and modulation of cell signaling pathways. These multifaceted mechanisms can be summarized as follows. 

### 3.1. Proteolytic Activity

Protein digestion: bromelain primarily acts as a proteolytic enzyme; this property allows bromelain to aid in protein digestion, facilitating the breakdown and absorption of dietary proteins in the gastrointestinal tract.

### 3.2. Fibrinolytic Activity

Breakdown of fibrin, a key component of thrombi and scar tissue: by promoting fibrinolysis, bromelain may help prevent excessive blood clot formation and improve circulation [33,34,35,36,37,38,39].

### 3.3. Antioxidant Effects

Scavenging free radicals: bromelain exhibits antioxidant properties by scavenging free radicals and reactive oxygen species (ROS). By neutralizing oxidative stress, bromelain helps protect cells and tissues from damage caused by oxidative injury [40,41,42,43,44,45,46,47,48,49,50,51]. 

### 3.4. Immune Modulation

Enhancement of immune function: bromelain exhibits immunomodulatory properties by enhancing various aspects of immune function. It promotes the activity of immune cells such as macrophages, natural killer (NK) cells, and lymphocytes, thereby enhancing immune surveillance and defense mechanisms against pathogens. Bromelain also helps maintain a balanced cytokine profile by modulating the production of both proinflammatory and anti-inflammatory cytokines. This balance is crucial for proper immune function and inflammatory responses [25,52,53,54,55,56,57,58,59,60,61,62,63,64]. 

In vitro experiments have shown that bromelain has a significant impact on various immune cells. Specifically, bromelain can modulate surface adhesion molecules on T cells, macrophages, and NK cells, which are crucial for cell–cell interactions and for the immune response in general. Furthermore, it can induce the secretion of proinflammatory cytokines by peripheral blood mononuclear cells (PBMCs), playing instrumental roles in inflammation and immune regulation. Bromelain inhibits the T cell signal transduction pathways, particularly the Raf-1/extracellular-regulated-kinase- (ERK-) 2 pathway; this inhibition disrupts the signaling processes necessary for T cell activation [65]. Consequently, treatment with bromelain results in decreased activation of CD4^+^ T cells, a subset of T cells essential for orchestrating the immune response, and reduces the expression of CD25, a marker of T cell activation [66].

Bromelain has been reported to have analgesic and anti-inflammatory effects; these effects are thought to stem from its ability to influence pain mediators directly, such as bradykinin [34], which is involved in the sensation of pain and the inflammatory response [67]. By modulating these pathways and mediators, bromelain demonstrates potential therapeutic benefits for conditions characterized by inflammation and pain.

### 3.5. Regulation of Specific Cell Signaling Pathways

PI3K/Akt and MAPK pathways: bromelain influences various cell signaling pathways involved in cell proliferation, survival, and inflammation. It modulates the phosphoinositide 3-kinase/protein kinase B (PI3K/Akt) pathway and the mitogen-activated protein kinase (MAPK) pathway, thereby regulating cellular responses to extracellular stimuli [55,56,57,61,68,69,70,71,72,73,74,75,76,77,78,79,80]. 

Studies in Sprague–Dawley rats demonstrated that the beneficial effects of bromelain are partly due to its ability to phosphorylate Akt, which, in turn, leads to the phosphorylation of FOXO3A [72]. 

Bromelain was also shown to exert inhibitory effects against LPS-stimulated inflammatory responses in RAW264.7 macrophage cells: the inhibition of iNOS and COX-2 expression by bromelain attenuated the production of IL-6 and TNF-α; the beneficial effects of bromelain on LPS-induced inflammatory responses were mainly associated with decreased expression of proteins involved in the NF-κB and MAPKs signaling pathways [56].

### 3.6. Down-Regulation of Plasma Kininogen

Bromelain has been shown to down-regulate plasma kininogen levels. Kininogens are precursors to kinins, which are potent mediators of inflammation and vasodilation [81,82]. By reducing plasma kininogen levels, bromelain may inhibit the production of kinins, thereby attenuating inflammation and its associated symptoms such as pain and swelling [34,81,82,83,84,85,86,87]. 

### 3.7. Inhibition of Prostaglandin E2 Expression

Prostaglandin E2 (PGE2) is a key mediator of inflammation and pain. It is synthesized from arachidonic acid by the enzyme cyclooxygenase (COX). Bromelain has been reported to inhibit the expression of COX enzymes, particularly COX-2, which are responsible for the synthesis of prostaglandins from arachidonic acid, thereby reducing the production of PGE2 [34,52,54,56,59,68,69,70,73,88,89,90,91]. By inhibiting PGE2 synthesis, bromelain helps mitigate inflammation and alleviate pain. 

### 3.8. Degradation of Advanced Glycation End Product Receptors

Advanced glycation end products (AGEs) are formed through nonenzymatic glycation and oxidation of proteins, lipids, and nucleic acids [61,92,93]. Accumulation of AGEs is associated with various pathological conditions, including diabetes, cardiovascular disease, and neurodegenerative disorders. Bromelain has been shown to degrade AGE receptors, such as the receptor for AGEs (RAGE), thereby reducing AGE-induced inflammation and tissue damage.

### 3.9. Regulation of Angiogenesis

Angiogenesis, the formation of new blood vessels from pre-existing ones, plays a crucial role in various physiological and pathological processes, including wound healing, tumor growth, and inflammation. Bromelain has been reported to regulate angiogenic biomarkers, including vascular endothelial growth factor (VEGF) and matrix metalloproteinases (MMPs) [59,72,94,95,96,97]. By modulating angiogenic factors, bromelain may influence vascular remodeling and tissue repair processes. 

The main biological activities and applications of fruit bromelain and stem bromelain are summarized in Table 2.

## 4. Medical Indications 

Numerous clinical trials have demonstrated the efficacy and safety of bromelain across various medical conditions, including osteoarthritis, sinusitis, surgical wounds, cardiovascular health, and digestive health [98,99,100,101,102,103,104,105,106,107,108,109,110,111,112,113,114,115,116,117,118]. However, larger-scale studies and further research are needed to confirm these findings and establish optimal dosing regimens and treatment protocols for different indications. Long-term safety data and potential drug interactions should be carefully evaluated in future studies. Currently, bromelain is considered to be nontoxic and without side effects. For instance, Castel and collaborators [119] demonstrated that the body can absorb a significant amount of bromelain, ~12 gm/day, without any major side effects. Other studies reported daily doses from 200 up to 2000 mg for prolonged periods of time and without concerns [28,120]. Notably, albeit bromelain has shown therapeutic benefit in doses as small as 160 mg/day, it is worth mentioning that the best results occur when starting at a dose of 750–1000 mg/day [18,28,120]. While bromelain is generally considered safe, some individuals may experience mild side effects, including gastrointestinal disturbances (nausea, diarrhea, and abdominal discomfort) and allergic reactions (including skin rashes, itching, and swelling of the lips, tongue, or throat). Bromelain may interact with certain medications, including some anticonvulsants, some antibiotics, and some anticoagulants (e.g., warfarin). 

The main medical conditions in which bromelain has been studied for its potential therapeutic effects are outlined in the following sections.

### 4.1. Inflammation, Edema, and Swelling

Bromelain is believed to have anti-inflammatory properties, which may help reduce swelling, pain, and inflammation associated with conditions such as arthritis, sports injuries, and postoperative recovery. Indeed, bromelain has been shown to inhibit the production of various inflammatory mediators, including cytokines, prostaglandins, and leukotrienes. These molecules play crucial roles in the initiation and propagation of inflammation. 

Bromelain exerts potent anti-inflammatory effects by modulating various inflammatory mediators, including cytokines, chemokines, and prostaglandins. It inhibits the production of proinflammatory cytokines such as interleukin-1 beta (IL-1β), tumor necrosis factor-alpha (TNF-α), and interleukin-6 (IL-6) [39,50,53,54,55,58,61,62,64,121,122,123,124]. 

Additionally, bromelain suppresses the nuclear factor-kappa B (NF-κB) signaling pathway, a key regulator of inflammation and immune responses; by inhibiting NF-κB activation, bromelain reduces the expression of inflammatory genes and attenuates the inflammatory cascade [56,57,125]. Bromelain’s proteolytic activity facilitates the breakdown of proteins involved in edema formation and swelling. By degrading extracellular matrix proteins and reducing the accumulation of fluid in tissues, bromelain may help alleviate inflammation-associated edema [113,126,127,128,129,130]. This property has led to the use of bromelain in various conditions characterized by edema and swelling, such as acute injuries, postoperative recovery, and inflammatory joint disorders like arthritis [131,132,133]. A recent clinical study has shown that the association of bromelain with vitamin C in postoperative bimalleolar surgery led to better outcomes, allowing a reduction in complications [134]. 

Notably, as mentioned above, bromelain has immunomodulatory effects that may influence the inflammatory response. It can modulate the activity of immune cells such as macrophages, lymphocytes, and dendritic cells, which play critical roles in initiating and regulating inflammation; actually, several studies suggest that bromelain may promote a shift towards anti-inflammatory immune responses by modulating the balance between proinflammatory and anti-inflammatory cytokines and chemokines [56,135,136,137,138,139,140]. All these beneficial actions have also suggested that bromelain could have a role in promoting tissue repair [106,141,142,143,144]. 

### 4.2. Digestive Health

Bromelain has been traditionally used to aid digestion and alleviate symptoms of indigestion, bloating, and heartburn. It is thought to assist in the breakdown of proteins in the digestive tract [18,51,145,146,147,148]. Bromelain has been shown to impact various aspects of digestion. Its proteolytic activity aids in the breakdown of dietary proteins into smaller peptides and amino acids. By hydrolyzing peptide bonds within protein molecules, bromelain facilitates their digestion and absorption in the gastrointestinal tract. This property of bromelain may be particularly beneficial for individuals with impaired protein digestion, such as those with pancreatic insufficiency or digestive enzyme deficiencies [147,149,150,151,152]. Through its proteolytic action, bromelain may enhance the bioavailability of nutrients derived from proteins. By breaking down protein complexes into simpler forms, bromelain may facilitate the absorption of essential amino acids and peptides across the intestinal mucosa. An improved protein digestion and nutrient absorption can contribute to overall nutritional status and support various physiological functions in the body. In addition, bromelain exhibits anti-inflammatory properties that may help alleviate digestive discomfort associated with inflammation in the gastrointestinal tract. By modulating inflammatory mediators and reducing mucosal inflammation, bromelain may mitigate symptoms such as bloating, gas, and abdominal pain. Numerous studies suggest that bromelain supplementation may be beneficial for individuals with inflammatory conditions of the digestive tract, including inflammatory bowel disease (IBD) and gastritis [10,25,62,141,153,154,155,156,157]. Bromelain may indirectly support the production and secretion of digestive enzymes by stimulating pancreatic and intestinal function. By promoting a favorable environment for enzymatic activity, bromelain may enhance overall digestive capacity [158]. Hence, bromelain’s anti-inflammatory effects may help protect pancreatic and intestinal tissues from damage, thereby preserving their functional integrity and ensuring optimal enzyme secretion. 

Henceforth, Bromelain’s proteolytic activity, anti-inflammatory properties, and potential modulation of gut microbiota collectively contribute to its beneficial effects on digestive health. Whether consumed as part of fresh pineapple or in supplement form, bromelain may support various aspects of digestion, including protein digestion, nutrient absorption, relief of digestive discomfort, and maintenance of gut microbiota balance. However, further research is needed to elucidate the specific mechanisms underlying bromelain’s effects on digestive health and its optimal therapeutic applications. 

### 4.3. Aging

The relationship between bromelain and aging represents an area of emerging interest, though direct studies specifically addressing this relationship are limited [159,160]. Cellular senescence is a state in which cells cease to divide but do not die, contributing to aging and the development of age-related diseases. Senescent cells exhibit a senescence-associated secretory phenotype (SASP), which includes the secretion of proinflammatory cytokines, chemokines, and proteases that can lead to tissue dysfunction and promote various diseases. Bromelain, a complex mixture of proteolytic enzymes derived from the pineapple plant, has been studied for its various biological activities, including anti-inflammatory, immunomodulatory, and potential anticancer effects. While direct research on bromelain’s impact on cellular senescence is not extensively documented, its known properties suggest several potential mechanisms through which it could influence senescence and the aging process, including anti-inflammatory effects, immunomodulatory properties, and proteolytic activity. Indeed, bromelain has been shown to modulate the inflammatory process, which is closely linked to the development and persistence of cellular senescence. By potentially reducing inflammation, bromelain could indirectly affect the senescence-associated secretory phenotype, thereby mitigating the adverse effects of senescent cells on tissue function [161,162,163,164]. Bromelain’s ability to modulate the immune response could also play a role in managing the effects of cellular senescence. By influencing immune cell activity, bromelain might help in clearing senescent cells, a process known as immunosurveillance, thus contributing to the maintenance of tissue homeostasis and reducing age-related pathologies. The proteolytic activity of bromelain could be relevant in the context of senescence by modulating the extracellular matrix and affecting the tissue microenvironment [165]. This activity might influence the behavior of senescent cells and their interaction with surrounding cells and matrix components [166]. Although the potential relationship between bromelain and cellular senescence presents an intriguing avenue for research, more targeted studies are needed to elucidate the mechanisms by which bromelain may influence senescence and aging. Understanding these mechanisms could open new therapeutic strategies for aging and senescence-associated diseases, leveraging bromelain’s bioactive properties.

### 4.4. Dermatology

Bromelain has found various applications in dermatology and cosmetics, leveraging its unique properties to address a range of skin concerns and enhance cosmetic procedures [167,168,169]. Bromelain’s proteolytic activity may contribute to wound healing and tissue repair processes by facilitating the removal of damaged tissue and promoting the proliferation of healthy cells [170,171]; by accelerating the resolution of inflammation and supporting the remodeling of injured tissues, bromelain may enhance the body’s ability to recover from injuries, surgical procedures, and other forms of tissue damage [96,106,108,171,172,173,174,175,176,177,178,179,180,181,182,183]. Bromelain is utilized to manage bruising and swelling following nonsurgical cosmetic procedures. Its application, either topically or orally, can lead to a reduction in the development of bruises and may also accelerate the healing process. This is particularly relevant following procedures that may lead to ecchymosis and edema, where bromelain’s anti-inflammatory properties can be beneficial. Bromelain acts as an exfoliating agent in cosmetic products. Its proteolytic activity enables it to gently remove dead skin cells, promoting skin renewal and improving texture. This enzymatic exfoliation is considered a gentler alternative to physical or chemical exfoliants, making it suitable for sensitive skin types. In the development of skin care products, such as cleansing washes and moisturizing lotions, bromelain is incorporated for its protease activity. This inclusion is based on its ability to aid in skin rejuvenation, enhance penetration of other active ingredients, and maintain skin hydration. Beyond cosmetic enhancements, bromelain’s properties have been explored for therapeutic applications in dermatology [168]. Its anti-inflammatory and wound-healing capabilities suggest potential benefits in treating various skin conditions, although more targeted research is needed to fully establish these therapeutic uses [184]. The use of bromelain, along with other botanical extracts like arnica, has been also indicated to minimize complications such as bruising and swelling following filler injections and other cosmetic interventions [185,186]; this application underscores bromelain’s role in post-procedure care, enhancing patient outcomes and satisfaction. Thus, as a botanical cosmeceutical, bromelain is recognized for its potential in improving skin health and appearance. Its incorporation into cosmetic products is supported by its stability and efficacy in various formulations, contributing to its growing popularity in the cosmetic industry.

### 4.5. Infectious Disorders

Some studies suggest that bromelain exhibits antibacterial properties [187,188,189], which may help prevent wound infections and promote a sterile wound environment conducive to healing. By inhibiting the growth of pathogenic bacteria and other microorganisms, bromelain may reduce the risk of wound complications and facilitate the healing process [95,178,189,190,191,192,193,194,195,196,197,198]. Bromelain has been recognized for its natural antimicrobial properties, which make it a candidate for treating infections of bacterial origin. Studies have highlighted its effectiveness against a variety of pathogens, suggesting its potential in managing infections, caries, and periodontal diseases among other conditions. The mechanism behind its antimicrobial activity may involve the disruption of bacterial cell walls or inhibition of bacterial adhesion and colonization. A specific case where bromelain has shown promise is in the treatment of *Pityriasis lichenoides chronica*, an infectious skin disease. Relatively recent observations indicate that bromelain treatment may lead to the complete resolution of this condition [199], showcasing its potential as a therapeutic agent in dermatological infectious disorders. Intriguingly, recent investigations have also proposed bromelain as a potential therapeutic strategy against COVID-19, caused by the SARS-CoV-2 virus [200,201]. Akhter and colleagues discovered that bromelain, when used alone at concentrations of 50 and 100 µg/mL, as well as in combination with acetylcysteine at concentrations of 50 and 100 µg/20 mg/mL, can disrupt the integrity of spike and envelope proteins of the SARS-CoV-2 virus [201]. Pretreatment with bromelain significantly hindered SARS-CoV-2 viral binding in VeroE6 cells, resulting in decreased viral infection and reduced SARS-CoV-2 viral RNA copies within the cells. Moreover, the combination of bromelain’s multifunctional enzymatic properties with acetylcysteine’s potent ability to break disulfide bonds led to inhibition of SARS-CoV-2 infectivity [201,202,203]. 

Ismail Celik and collaborators conducted a study utilizing molecular docking and molecular dynamics simulation techniques to explore the potential of bromelain in combating different variants of SARS-CoV-2 by targeting its interaction with angiotensin converting enzyme 2 (ACE2); the researchers discovered that bromelain demonstrated favorable binding affinity to various variants of the receptor-binding domain (RBD), which is crucial for the binding of the virus to ACE2 [204]. However, further studies are needed to validate these findings before any definitive conclusions can be drawn. The exploration into bromelain’s binding interactions with components of the virus suggests that it could be helpful in the management of COVID-19 as well as other bacterial-mediated diseases. This aspect is particularly relevant given the ongoing global health crisis and the need for effective treatments against the virus. Bromelain’s effects on the immune system are also pertinent to its role in infectious disorders. It has been shown to induce macrophage apoptosis and activation, which are crucial processes in the immune response to infections. By modulating the immune system, bromelain can help in treating immune-mediated conditions and potentially enhance the body’s defense against infections. Lastly, the development of bromelain-capped gold nanoparticles has been recommended as a novel drug delivery system for treating infectious diseases, including catheter-associated urinary tract infections. This innovative approach highlights the versatility of bromelain in medical applications and its potential to improve the efficacy of existing treatments.

### 4.6. Cancer

The relationship between bromelain and cancer has been a subject of increasing interest within the scientific community, particularly in exploring its potential as a therapeutic agent. In fact, there is emerging evidence suggesting anticancer properties of bromelain, with studies indicating potential benefits in inhibiting tumor growth, preventing metastasis, and enhancing the effectiveness of chemotherapy [17,18,51,205,206,207,208,209,210,211,212,213,214,215]. While research is ongoing and more evidence is needed, several mechanisms have been proposed to explain bromelain’s effects on cancer cells, including inhibition of cancer cell proliferation and induction of apoptosis, antiangiogenic effects, modulation of inflammatory pathways, enhancement of immune function, as well as by increasing the sensitivity of cancer cells to therapeutic interventions [94,141,195,197,198,199,200]. A novel approach involving the combination of bromelain and acetylcysteine has been recently discussed for its implications in cancer therapy [200]. This combination targets mucins in cancer cells, which are involved in the progression and metastasis of cancer. Mounting evidence suggests potential for bromelain + acetylcysteine in enhancing the efficacy of chemotherapy [76,216,217], although further studies are needed to fully understand its therapeutic potential and mechanisms of action. In vitro assays have demonstrated that bromelain can induce apoptosis in cancer cells, including breast cancer cells (specifically GI-101A cells) [218]. This phenomenon suggests that bromelain may contribute to the inhibition of cancer cell growth and potentially enhance the effectiveness of conventional cancer treatments. The cytotoxic effects of both unfractionated and fractionated bromelain on colorectal cancer cells have been investigated, alone or in combination with chemotherapeutic agents [214]; the findings indicate that bromelain treatment results in reduced cell survival in colorectal cancer cells in a dose-dependent manner, highlighting its potential as a complementary therapy in colorectal cancer treatment. Several studies have also explored the ACE2-inhibitory effects of bromelain in colon cancer cells [219]. Given the role of ACE2 in various physiological processes and its implications in cancer, bromelain’s inhibitory effects could offer a novel approach to targeting cancer cells [220], although the specific mechanisms and clinical relevance require further investigation.

## 5. Conclusions

Bromelain is a natural enzyme complex with diverse potential therapeutic effects, ranging from anti-inflammatory and digestive properties to immune modulation, aging, and wound healing. While research on bromelain continues to expand, further well-designed clinical trials are needed to elucidate its mechanisms of action, optimal dosing regimens, and efficacy in various medical conditions. Since its anti-inflammatory and antioxidant activities could be synergic with vitamin C, a possible association with this ingredient could be of interest. Furthermore, as illustrated above, anti-inflammatory and antioxidant activities can be achieved by combining 1000 mg bromelain with 500 mg of vitamin C. Healthcare providers should be aware of potential side effects and drug interactions associated with bromelain supplementation to ensure safe and appropriate use in clinical practice. 

## Figures and Tables

**Figure 1 nutrients-16-02060-f001:**
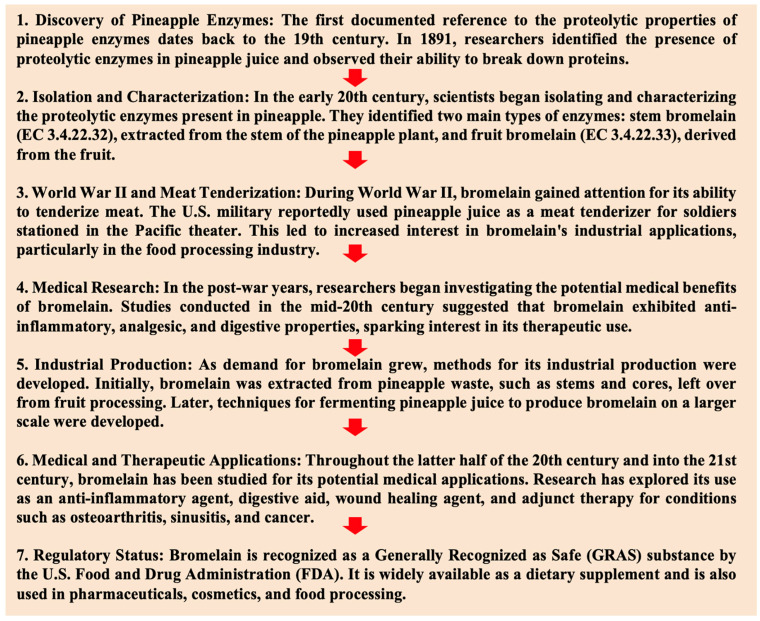
Timeline highlighting the key events in the discovery and development of bromelain.

**Table 1 nutrients-16-02060-t001:** Summary of the main features of fruit bromelain and stem bromelain.

	*Fruit Bromelain* (EC 3.4.22.33)	*Stem Bromelain* (EC 3.4.22.32)
** *Source and extraction* **	It is primarily extracted from the fruit (particularly the core) of the pineapple plant. It is obtained by crushing or juicing the fruit and then separating the bromelain enzyme from other components.	It is extracted from the stems of the pineapple plant. The stems contain a higher concentration of bromelain compared to the fruit, and the extraction process involves grinding or macerating the stems to release the enzyme.
** *Composition* **	Typically contains a mix of proteolytic enzymes, including various cysteine proteases, such as stem bromelain, ananain, and comosain. It may also contain other enzymes and bioactive compound.	It consists mainly of cysteine proteases, with the predominant enzyme being bromelain. It may also contain trace amounts of other proteolytic enzymes.
** *Enzymatic action* **	Hydrolysis of proteins with broad specificity for peptide bonds. Bz-Phe-Val-Arg-/-NHMec is a good synthetic substrate, but there is no action on Z-Arg-Arg-NHMec.	Broad specificity for cleavage of proteins but strong preference for Z-Arg-Arg-/-NHMec amongst small molecule substrates.

**Table 2 nutrients-16-02060-t002:** Biological activities and main applications of fruit bromelain and stem bromelain.

	*Fruit Bromelain* (EC 3.4.22.33)	*Stem Bromelain* (EC 3.4.22.32)
*Biological activities*	Proteolytic activity Anti-inflammatory effects Immunomodulatory effects Antioxidant effects	Enzymatic properties Anti-inflammatory effects Immunomodulatory effects Antioxidant effects Promotion of wound healing It may also aid in digestion
*Main applications*	Food processing (e.g., meat tenderizer) Alternative medicine practices Skincare products	Food processing (e.g., meat tenderizer) Pharmaceuticals Cosmetics It is often preferred for its higher enzymatic activity and purity.

## Data Availability

Not applicable.

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
