# Peer review of "Exploring the Therapeutic Potential of Bromelain: Applications, Benefits, and Mechanisms"

_nutrients, 2024, doi:10.3390/nu16132060_

Round 1

Reviewer 1 Report

Comments and Suggestions for Authors

Dear all,

In this form the current paper cannot be accepted for publication ... A good review always has some good figures, tables, etc...

Then... you name it an update ... ok, from when ? this has to be clear ... the review covers which period ? the novelty has to be highlighted ... what is different from previous works ... etc...

Author Response

Dear all,

In this form the current paper cannot be accepted for publication ... A good review always has some good figures, tables, etc...

Then... you name it an update ... ok, from when ? this has to be clear ... the review covers which period ? the novelty has to be highlighted ... what is different from previous works ... etc...

RE: We thank this Reviewer for her/his comments. Our Review has 3 items (Figures/Tables) and 220 References. Therefore, we respectfully disagree with this Reviewer. Additionally, both Reviewer #2 and #3 agreed with our view.

Our review covers studies published until 2024; we removed the word "update". from the title in order to avoid any confusion to the Readers on this aspect.

Reviewer 2 Report

Comments and Suggestions for Authors The title of the work corresponds to its content. The work was prepared correctly and reliably, the authors reviewed the activity of bromelains. The only thing missing in the study was information about potential side effects if the therapeutic dose was exceeded. I believe that such data should be included.
References are relevant to the work and include the latest news in this field.

Author Response

The title of the work corresponds to its content.

The work was prepared correctly and reliably, the authors reviewed the activity of bromelain.

The only thing missing in the study was information about potential side effects if the therapeutic dose was exceeded. I believe that such data should be included.

References are relevant to the work and include the latest news in this field.

RE: We thank this Reviewer for her/his pertinent comments.

We have added a section to discuss the potential side effects, as requested.

Reviewer 3 Report

Comments and Suggestions for Authors

In their manuscript "Bromelain and Human Health: An Update”, Urna Kansakar and coworkers wrote about the history, structure, chemical properties, and medical indications of bromelain. The clinical and medicinal interest in bromelain is growing, and the update is useful to suggest its evidence-based application. 

In this paper, the bibliography is adequate, but in my opinion, there are some aspects that should be considered before the paper publication. 

Major points: 

1. In table 1, please add that the data reported were adapted from BRENDA:EC3.4.22.32 and BRENDA:EC3.4.22.33 or rewrite the text. 

2.  In the section 3, the contents of paragraphs are very superficial. I advise to be more specific, writing the details of biological activity. For example, questions that should be addressed are: (i) what are the molecular target of bromelain in the different biological activities? (ii) a specific cell signaling pathways, in which way is it modulated?  

3. Moreover, authors wrote that Bromelain regulates “cellular responses to extracellular stimuli” by action on PI3K/Akt and MAPK Pathways. Also in this case, major details could be added.   

4. In the conclusion the authors wrote “Since its anti-inflammatory and antioxidant activities could be synergic with vitamin C, a possible association with this ingredient could be of interest. Furthermore, as illustrated above, anti-inflammatory and antioxidant activities can be achieved by combining 1000 mg bromelain with 500 mg of vitamin C.” In the main text, there are not data about it. So, I suggest to discuss the combination of bromelain with of vitamin C with more detail in the main text.

So, in general, the manuscript should be revised by eliminating redundant information and by adding more details.

Comments on the Quality of English Language

Good

Author Response

In their manuscript "Bromelain and Human Health: An Update”, Urna Kansakar and coworkers wrote about the history, structure, chemical properties, and medical indications of bromelain. The clinical and medicinal interest in bromelain is growing, and the update is useful to suggest its evidence-based application. 

In this paper, the bibliography is adequate, but in my opinion, there are some aspects that should be considered before the paper publication. 

Major points: 

  1. In table 1, please add that the data reported were adapted from BRENDA:EC3.4.22.32and BRENDA:EC3.4.22.33 or rewrite the text. 
  2.  In the section 3, the contents of paragraphs are very superficial. I advise to be more specific, writing the details of biological activity. For example, questions that should be addressed are: (i) what are the molecular target of bromelainin the different biological activities? (ii) a specific cell signaling pathways, in which way is it modulated?  
  3. Moreover, authors wrote that Bromelain regulates “cellular responses to extracellular stimuli” by action on PI3K/Akt and MAPK Pathways. Also in this case, major details could be added. 
  4. In the conclusion the authors wrote “Since its anti-inflammatory and antioxidant activities could be synergic with vitamin C, a possible association with this ingredient could be of interest. Furthermore, as illustrated above, anti-inflammatory and antioxidant activities can be achieved by combining 1000 mg bromelain with 500 mg of vitamin C.” In the main text, there are not data about it. So, I suggest to discuss the combination of bromelain with of vitamin C with more detail in the main text.

So, in general, the manuscript should be revised by eliminating redundant information and by adding more details.

RE: We thank this Reviewer for her/his comments.

1. We now specify that the Data in Table 1 were adapted from BRENDA, as recommended by this Reviewer.

2-3. We agree with this Reviewer and we have expanded section 3, as recommended, in order to include the details of molecular targets and cell signaling pathways, with references.

4. We thank this Reviewer for this comments. In the main text we have added data from recent observations (Meccariello et al.) on the association of Vitamin C and Bromelain; we apologize for the oversight.

We have revised our manuscript in order to eliminate redundant information and by adding more details, as recommended.